# Experimental Study on Bead on Plate (BOP) Welding of 6 mm Thick 9% Nickel Steel by Fiber Laser Welding

**DOI:** 10.3390/ma14247699

**Published:** 2021-12-13

**Authors:** Jaewoong Kim, Changmin Pyo, Yonghyun Kim, Sungwook Kang, Taegon Yeo, Kwangsan Chun, Du-Song Kim

**Affiliations:** 1Automotive Materials & Components R&D Group, Korea Institute of Industrial Technology, Gwangju 61012, Korea; kjw0607@kitech.re.kr (J.K.); changmin@kitech.re.kr (C.P.); kyh1927@kitech.re.kr (Y.K.); 2Precision Mechanical Process and Control R&D Group, Korea Institute of Industrial Technology, Jinju 52845, Korea; swkang@kitech.re.kr; 3Teajin Engineering, 13, Jayumuyeok-ro Samho-eup, Yeongam-gun 58453, Korea; taejin0093@naver.com; 4Industrial Application R&D Group, Welding Engineering R&D Department, Daewoo Shipbuilding & Marine Engineering Co., Ltd., Geoje-si 53302, Korea; 5Department of Welding and Joining Science Engineering, Graduate School, Chosun University, 309 Pilmun-daero, Dong-Gu, Gwangju 501759, Korea

**Keywords:** nine percent nickel steel, fiber laser welding, LNG-fueled tank, bead on plate

## Abstract

Nine percent nickel steel has excellent properties in a cryogenic environment, so it has recently been used as a tank material for most LNG fuel-powered ships. However, 9% nickel steel causes arc deflection due to its tendency of magnetization during manual FCAW welding and the currently used filler metal is 10–25 times more expensive as a base metal compared to other materials, depending on manufacturers. Furthermore, the properties of its filler metal cause limitation in the welding position. To overcome these disadvantages, in this study, the tendency of penetration shape was analyzed through a fiber laser Bead on Plate (BOP) welding for 9% nickel steel with a thickness of 6 mm and a range of welding conditions for 1-pass laser butt welding of 6 mm thick 9% nickel steel with I-Groove were derived. Through this study, basic data capable of deriving optimal conditions for laser butt welding of 9% nickel steel with a thickness of 6 mm were obtained.

## 1. Introduction

As environmental pollution and global warming issues come to the fore, the demand for eco-friendly energy is increasing and environmental regulations are becoming stricter. In this environment, natural gas (NG) is in the spotlight because it emits a lower volume of pollutants than petroleum [1,2]. Since the NG production area is very limited around the world, it is transported in the form of liquefied natural gas (LNG) after liquefaction. Transportation is enabled by LNG carriers with LNG tanks and onshore LNG tanks in which LNG can be stored. Recently, there has been continuous development in LNG-related industries such as LFS (LNG-Fueled Ship) using LNG as fuel and LNG bunkering for fuel injection [2].

LNG has a cryogenic temperature of −163 degrees Celsius and there is a risk of explosion in the event of a leak, so extreme care is required during handling. Under the condition of −163 degrees Celsius, other steel materials including SS400 cannot be used due to brittleness and thus metals without low-temperature brittleness must be used. In its IGC Code, the International Maritime Organization (IMO) defines 9% Nickel steel, STS304L, 36% Nickel steel, AL5083 and high manganese steel as metals that can be used at cryogenic temperatures [3]. The material for LNG fuel tanks ordered recently is 9% nickel steel, and most of them are manufactured by FCAW (Flux-cored Arc Welding) welding. In this study, basic research was conducted to apply laser welding to overcome the FCAW problems of 9% nickel steel materials (quality degradation due to magnetization, expensive filler metal, difficulty in automation, etc.).

The most important processing technology to produce an LNG storage tank using the above materials is welding. Welding, one of the main production processes, is the most widely used technology to produce parts and finished goods in a range of industries including automobile, aviation, shipbuilding, etc. Conventionally, arc welding has been widely used due to its low operating cost and accessibility but has a number of associated problems such as thermos-elastic deformation because a large area is affected by high heat input [4]. On the other hand, laser welding can minimize thermos-elastic deformation, a disadvantage of conventional arc welding, by applying a concentrated heat source to a narrow area for a short time and also enables productivity improvement through its relatively fast welding speed [5,6,7,8]. Based on these advantages, laser welding is being introduced into the industrial fields and diverse research is being actively performed to derive welding techniques and optimum welding parameters for high welding quality and low welding distortion [9,10].

Due to the above advantages, fiber laser welding is widely used for cryogenic materials, and related research is being continuously conducted. Wu et al. conducted a study on spatter during fiber laser welding of Al5083, a material for cryogenic use [11,12] and Fang et al. performed welding of stainless steel using a 20 kw fiber laser and analyzed the results [13]. Pang et al. conducted a study on the microstructure and mechanical properties of the aluminum alloy during fiber laser welding [14] and Jiang et al. analyzed the porosity due to welding and conducted a study on the porosity defect caused by it [15]. Xin et al. confirmed the cryogenic impact toughness of AISI304L material during fiber laser welding [16] and Kim et al. studied the post-welding quality of thick high manganese steel through laser-MIG (Metal Electrode Inert Gas) hybrid welding [17]. As such, research in the industrial field and academia is continuously being carried out to replace the method using fiber laser instead of MIG welding as a welding method for cryogenic materials. In order to make an LNG cargo containment system, cryogenic materials must be used, and in order to manufacture using cryogenic materials, the demand for fiber lasers with less welding deformation is increasing.

Of the cryogenic materials mentioned above, 9% nickel steel was selected for further research in this study. The 9% nickel steel has excellent mechanical properties at the −163 degrees Celsius temperature required for LNG [3] such as yield strength and tensile strength, so it is widely used as a material for fuel propulsion tanks of LNG-fueled ships. Accordingly, there have been many studies on welding using 9% nickel steel.

Huang et al. conducted a study using filler metal for fiber laser welding of 9% nickel steel [18] and Choi et al. performed a study on 9% nickel steel as a material for type B LNG fuel tank [19]. Park et al. performed and analyzed Super TIG (Tungsten Inert Gas) welding to weld 9% nickel steel [20]. Na et al. studied the characteristics of 9% nickel steel welding by applying GTAW (Gas Tungsten Arc Welding) and MIG [21] and Yun et al. conducted research on the optimal welding method for fillet laser welding [22]. Kim et al. also studied the design of LNG-fueled ships using 9% nickel steel [23].

However, due to the arc deflection caused by magnetization during manual FCAW welding and the filler metal that is 10–25 times more expensive than other materials [24,25], there are many difficulties when 9% nickel steel is applied in the field, despite its excellent material performance. This study sought to solve the above problems by studying automatic welding using the fiber laser welding method. To identify the welding conditions that satisfy the welding performance in 1-pass laser butt welding of 6 mm thick 9% nickel steel, the range of welding conditions was explored using the design of experiments (DOE) after analyzing the tendency through Bead on Plate (BOP) welding.

## 2. Materials and Methods

### 2.1. Materials of Fiber Laser BOP Welding

The material used in this experiment is 6 mm thick 9% nickel steel produced by Nippon Steel, Japan and is ASTM A553M-17 with QT (Quenched-Tempered) treatment. The chemical composition and mechanical properties described in the Inspection Certification provided by the material manufacturer are shown in Table 1 and Table 2 below.

The size of the material used in the experiment is 300 mm × 300 mm × 6 mm, as shown in Figure 1 below and a total of 8 BOP welding were performed on one plate. As shown in the figures below, the interval between BOP welding is 60 mm and the length of each BOP welding is 240 mm. The next BOP welding was performed after the surface temperature of a previous BOP welding was sufficiently cooled down to 30 degrees or less, so that each BOP welding did not affect the subsequent experiments.

### 2.2. Methods of Fiber Laser BOP Welding

For BOP welding, 5 kW fiber laser weld equipment from Miyachi, Japan was used. The optical system used in this study has a spot diameter of 400 μm, a focal length of 148.8 mm and a focal depth of 6 mm. The 6-axis automatic robot was from Yaskawa, Japan and enables a constant focus position during laser welding. They are shown as Figure 2. The shielding gas was 99.99% Nitrogen with a speed of 15 L/min. Both the tilting angle and working angle were set to 0 degrees for the experiment.

After the BOP welding, a specimen was collected at a position of 120 mm, which is in the middle of each BOP welding length of 240 mm and a cross-sectional observation was performed. After cross-sectional observation, the penetration size was measured for each condition. Five measurements were taken: Top HAZ (Heat Affected Zone) width, top bead width, bottom HAZ width, HAZ depth and penetration. They are shown as Figure 3.

The cross-sectional measurement positions of the melted zone and the heat-affected zone are as follows:

(1) Top HAZ width: length of the HAZ of the base material surface (Top) observed through the micro-section

(2) Top bead width: melted surface width of base material that can be observed with naked eyes

(3) Bottom HAZ width: width of the melting zone at the endpoint of the penetration depth

(4) HAZ depth: vertical depth of HAZ from the surface of base metal

(5) Penetration: vertical depth of melting zone from the surface of base metal

For the BOP welding, a total of 28 experiments were performed with laser power and BOP welding speed as the variables. The range of variables in the experiment is shown in Table 3 below.

## 3. Results

### 3.1. Results of BOT Test

The results of penetration measurement through cross-sectional observation after the BOP test are shown in Table 4 below. The first part of each number means laser power and the rear number is listed in ascending order of speed for each corresponding laser power.

For HAZ depth and penetration in each experiment, the maximum value of each factor was 6 mm and it is expressed as 6 (PbT) in the table since the thickness of the material was 6 mm. The cross-sectional observation results in each experiment with a 3 mm scale bar are shown in Figure 4. All cross-sectional observation results were tested in accordance with ASTM E340 and the test was performed at an accredited certification institute. For the cross-sectional observation experiment, sandpaper polishing was performed up to 2000 times and polishing was performed up to 1 um. A solution of 100 mL of alcohol and 10 mL of 35% nitric acid was used as the etching solution. The microscope magnification was fixed at 10 times.

In Figure 4, it can be seen that the bead size is mostly increased as the welding speed is slower and the output is higher. Additionally, when speed is slow in the figure, it can be seen that the difference between the size of HAZ and the size of penetration is large. In both areas, different shades can be seen in cross-sectional observation due to the phase transformation of the material after laser welding. In the case of HAZ, it is not melted but means a portion where phase transformation has occurred due to heat, and penetration is a completely melted portion, and it can be seen that it appears narrower than the area of HAZ. In addition, it is shown that there are lower bead defects when the welding speed is slow, in cross-sectional observation cases 2-1, 3-1, 4-1 and 5-1. To secure the 1-pass welding condition of 6 mm thick 9% nickel steel, which is the purpose of this experiment, it is necessary to ensure that the lower bead is not defective, among the conditions where the output and speed conditions cause full penetration.

### 3.2. Analysis of BOP Welding Results

To analyze the BOP welding measurement results, the effect of each factor on penetration was analyzed. Figure 5 and Figure 6 show the bead results according to the welding speed under each laser output condition in the BOP welding.

In Figure 5, it can be seen that the results of top HAZ width and top bead width become smaller as the speed increases, regardless of the laser output condition. In the graph, it can be seen that in general, the decrease is large under the conditions of 0.25 mpm~1.00 mpm(meter per minute) and is small under the condition of 1.00 mpm or more. Additionally, it can be seen that the difference between top HAZ width and Top bead width is maintained almost uniformly under the condition of 1.50 mpm or more.

In Figure 6, the red dotted line at the position 6 mm on the *Y*-axis is the thickness of material used and HAZ depth and penetration indicate values within 6 mm. When the result is 6 mm, a value of 6 mm or more can be obtained, if the thickness of the material used is thicker than 6 mm. As the figure shows, the results of HAZ depth and penetration become smaller as the speed increases, regardless of the laser output. Furthermore, the difference between the HAZ depth and penetration results decreases as the speed increases.

## 4. Discussion

Figure 7 shows the comparison of HAZ depth and penetration values for each output. From this figure, the candidate conditions for 1-pass welding of 6 mm thick 9% nickel steel can be derived.

For 1-pass welding, it is necessary to secure the conditions of HAZ depth and penetration close to 6 mm and the lower bead free of defects. The condition range for 1-pass welding that satisfies this in the penetration result is as Table 5. In actual butt welding, a fine gap may exist. By considering this fact, the condition in which penetration and HAZ depth of 5 mm or more are achieved and there is no defect in the lower bead visible in the cross-sectional observation result is chosen.

Depending on the laser welding specification, the output may exceed 5 kW and the thickness of the same material may be different. In this case, it is necessary to determine the approximate range of welding experiments for 1-pass welding. In this experiment, the penetration value is usually proportional to the welding output and is inversely proportional to the welding speed. From the energy density (kW/mpm), i.e., welding output divided by welding speed, the range of welding conditions for full penetration can be configured when laser welding output specifications are different and when laser welding is performed on 9% nickel steel of 6 mm or less. Figure 8 shows the HAZ depth and penetration values according to the energy density. For example, for complete welding of 5 mm 9% nickel steel, the welding experiment should be performed under the condition of an energy density between 1.75 and 3.00.

## 5. Conclusions

In this study, a basic study was conducted to replace the existing FCAW welding of 9% nickel steel, which is used as the main material of the LNG fuel tank, with fiber laser welding. The 28 cases of BOP welding were performed to secure 1-pass butt welding conditions of 9% nickel steel with a thickness of 6 mm and a cross-sectional observation and analysis were performed.

(1)The BOP welding was performed using laser power and BOP welding speed as factors, and the laser power range was 2.00~5.00 Kw, and the BOP welding speed range was 0.25~3.00 mpm, for a total of 28 cases. From the cross-sectional observation results using ASTM E340, it was confirmed that, in general, the slower the welding speed and the higher the output and the larger the bead size. In addition, it was confirmed that the lower bead defect may appear at a certain speed or less.(2)For the four bead shapes (top bead width, top HAZ width, HAZ depth, penetration), the trend of bead size was analyzed by welding speed and output, and through this, it was possible to derive the optimal condition candidates for 1-pass butt welding of 9% nickel steel with a thickness of 6 mm.(3)In addition, the energy density factor was proposed by utilizing this experimental data and from this, the range of experimental conditions for 1-pass full penetration of 9% nickel steel under different laser welding conditions (thickness change, output change) could be configured.(4)Based on this study, in the future, it is planned to conduct a study to secure the mechanical properties of laser butt welding. Through this, it is expected that it will be possible to prove that the existing FCAW welding can be replaced with laser welding when manufacturing an LNG fuel-propelled tank made of 9% nickel steel in the current ship-building industry.

## Figures and Tables

**Figure 1 materials-14-07699-f001:**
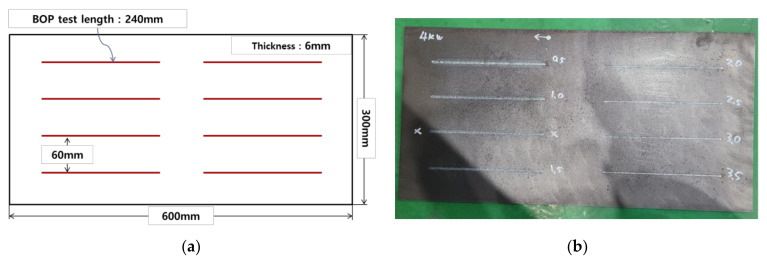
(**a**) BOP welding location on a specimen (**b**) Actual test sample.

**Figure 2 materials-14-07699-f002:**
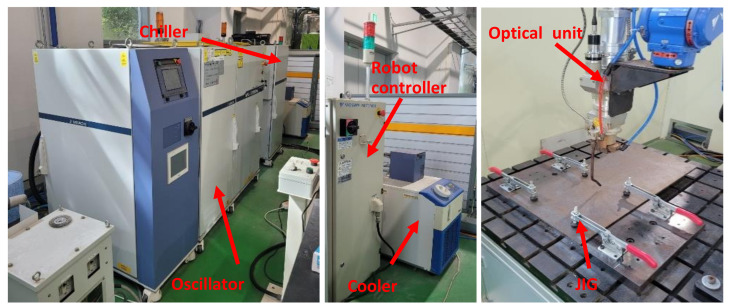
Fiber laser welding system.

**Figure 3 materials-14-07699-f003:**
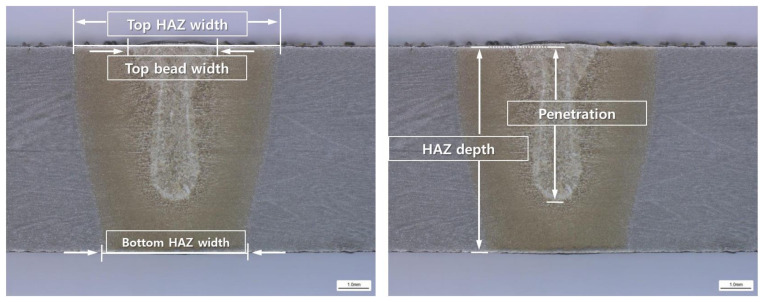
Cross-section measurement location and definition.

**Figure 4 materials-14-07699-f004:**
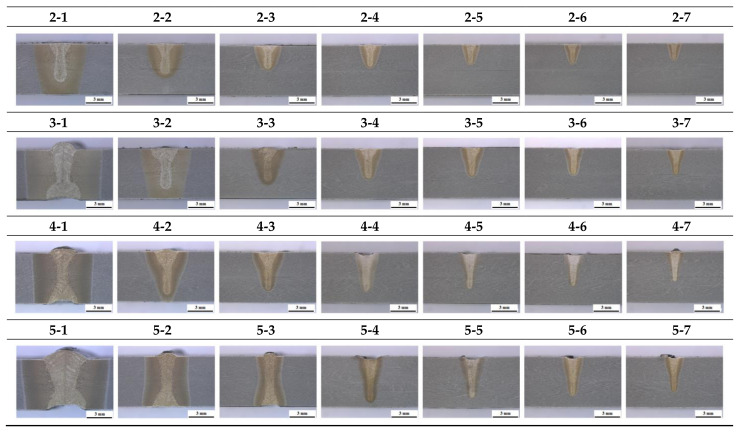
Cross-sectional observation results for each experiment.

**Figure 5 materials-14-07699-f005:**
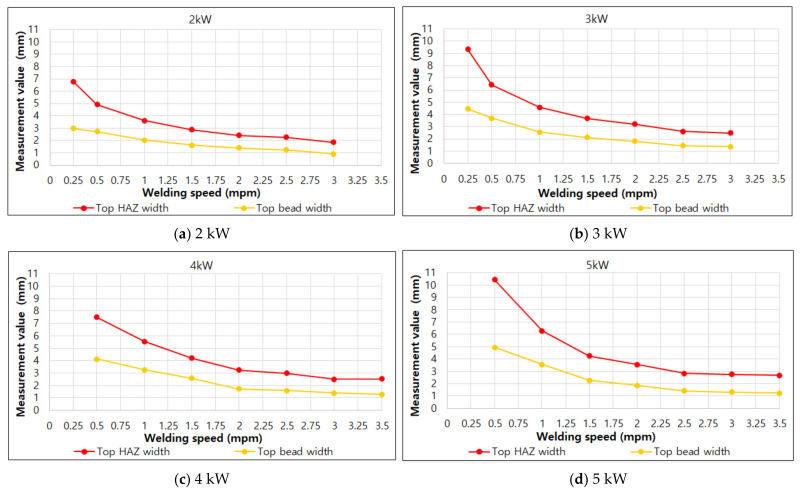
Results of top HAZ width and top bead width according to welding speed under each laser output condition.

**Figure 6 materials-14-07699-f006:**
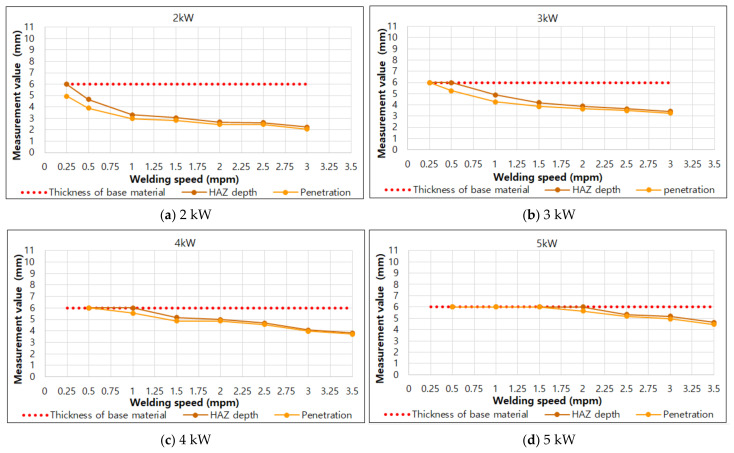
HAZ depth and penetration according to welding speed under each laser output condition.

**Figure 7 materials-14-07699-f007:**
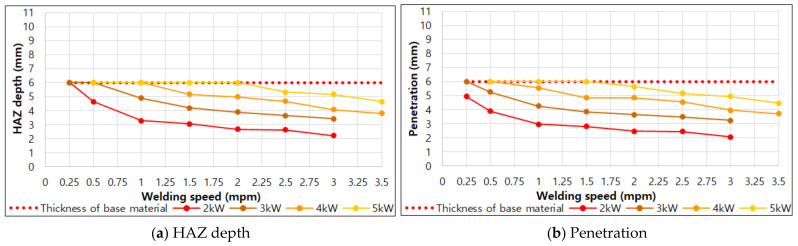
HAZ depth and penetration according to welding speed under each laser output condition.

**Figure 8 materials-14-07699-f008:**
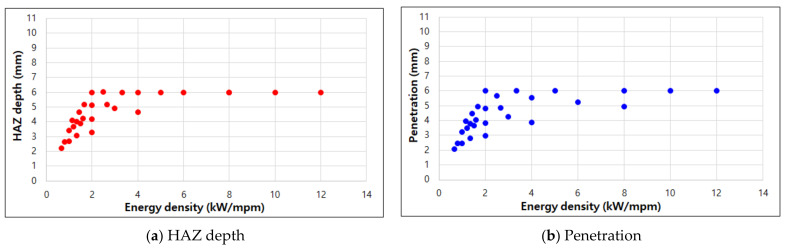
Relationship between energy density and HAZ depth, penetration.

**Table 1 materials-14-07699-t001:** Chemical composition of 9% nickel steel.

Component	Chemical Composition (wt.%)
Carbon, C	0.15
Manganese, Mn	0.64
Silicon, Si	0.23
Phosphorous, P	0.003
Sulfur, S	0.001
Nickel, Ni	8.91

**Table 2 materials-14-07699-t002:** Mechanical properties of 9% nickel steel.

Mechanical Property	Value
Yield strength	691 MPa
Tensile strength	728 MPa
Elongation	32.2%

**Table 3 materials-14-07699-t003:** Range of experimental BOP welding conditions.

Parameter of BOP Welding	Ranges of BOP Welding Conditions
Laser power (kW)	2.00~5.00
BOP welding speed (mpm, meter per minute)	0.25~3.00

**Table 4 materials-14-07699-t004:** Results of BOP welding.

#	Laser Power	Speed	Top HAZ Width (mm)	Top Bead Width (mm)	Bottom HAZ Width (mm)	HAZ Depth (mm)	Penetration (mm)
2-1	2	0.25	6.748	2.971	4.613	6 (PbT) ^1^^)^	4.957
2-2	2	0.50	4.927	2.717	-	4.658	3.897
2-3	2	1.00	3.613	2.045	-	3.300	2.971
2-4	2	1.50	2.881	1.627	-	3.061	2.822
2-5	2	2.00	2.404	1.403	-	2.672	2.478
2-6	2	2.50	2.254	1.239	-	2.628	2.463
2-7	2	3.00	1.851	0.911	-	2.225	2.060
3-1	3	0.25	9.346	4.449	10.003	6 (PbT) ^1^^)^	6 (PbT) ^1^^)^
3-2	3	0.50	6.435	3.703	4.360	6 (PbT) ^1^^)^	5.255
3-3	3	1.00	4.584	2.568	-	4.897	4.270
3-4	3	1.50	3.673	2.135	-	4.195	3.852
3-5	3	2.00	3.210	1.807	-	3.897	3.658
3-6	3	2.50	2.613	1.448	-	3.658	3.494
3-7	3	3.00	2.478	1.374	-	3.434	3.255
4-1	4	0.50	7.510	4.136	8.182	6 (PbT) ^1^^)^	6 (PbT) ^1^^)^
4-2	4	1.00	5.554	3.270	1.403	6 (PbT) ^1^^)^	5.554
4-3	4	1.50	4.210	2.583	-	5.181	4.867
4-4	4	2.00	3.240	1.717	-	5.000	4.860
4-5	4	2.50	3.001	1.598	-	4.677	4.567
4-6	4	3.00	2.508	1.388	-	4.091	3.976
4-7	4	3.50	2.538	1.284	-	3.823	3.726
5-1	5	0.50	10.421	4.942	11.735	6 (PbT) ^1^^)^	6 (PbT) ^1^^)^
5-2	5	1.00	6.300	3.553	5.748	6 (PbT) ^1^^)^	6 (PbT) ^1^^)^
5-3	5	1.50	4.255	2.284	3.628	6 (PbT) ^1^^)^	6 (PbT) ^1^^)^
5-4	5	2.00	3.553	1.866	-	6.012	5.640
5-5	5	2.50	2.837	1.418	-	5.329	5.174
5-6	5	3.00	2.762	1.329	-	5.181	4.942
5-7	5	3.50	2.672	1.239	-	4.643	4.464

(PbT) ^1^^)^: Penetration by thickness.

**Table 5 materials-14-07699-t005:** Condition range for 1-pass welding of 6 mm 9% nickel steel.

Laser Power (kW)	Range of Welding Speed (mpm)
2	0.25
3	0.25~0.50
4	0.50~1.00
5	1.00~2.00

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
