# Peer review of "Experimental Study on Bead on Plate (BOP) Welding of 6 mm Thick 9% Nickel Steel by Fiber Laser Welding"

_materials, 2021, doi:10.3390/ma14247699_

Round 1

Reviewer 1 Report

Review report on the topic ‘Experimental Study on Bead on Plate of 6 mm Thick 9% Nickel  Steel for LNG Fueled Tank by Fiber Laser Welding’.

The suggestion to improve the quality of the manuscript is listed below:
1. The novelty of the work is not clear. Provide a separate section.
2. The application of work should be discussed first.
3. The abstract section is very lengthy. Remove the unnecessary information and add the 
key finding of the works.
4. The introduction section is very weak. Try to add a few current published works and 
make a bridge between previously published works with current work. 
5. Provide the proper welding parameters and also include the heat input.
6. How was the composition of the base plate analyzed?
7. Provide the image of the top and bottom view of the welded plate.
8. The variation in bead width and depth is ok, but the technical discussion is missing.
9. Provide the key bullet points in the conclusion section. 

Reviewer 2 Report

The article deals with experimental BOP tests on a laser welding machine.
The authors present the results of penetration tests.

The title of the article evokes that the results of the article should be related to the use in LNG Fueled Tanks.
LNG appears 19 times in the article itself, but the last mention is at the end of the introduction.

Therefore, in relation to the chosen title, the article lacks any discussion and does not contain any scientific conclusions.

Thus, the authors did not manage to define the novelty and contribution of the article.

Therefore, in my opinion, the article does not meet the minimum requirements for publications in the Materials journal.

Reviewer 3 Report

This paper presents an experimental study on the tendency of penetration shape of nickel steel under fiber laser welding. The experiments were designed relatively well and the presentation of the manuscript is easy to follow. However, the depth of the discussion can be further improved and the key scientific innovations should be highlighted. The following comments should be addressed.

  1. The scale bars of Figure 3 are not clear. Scale bars should be added for Figure 4.
  2. Some of the figures with similar contents, e.g. Figures 5-8 can be merged.
  3. Mechanisms of the different HAZ depth and penetrations should be presented; otherwise, the paper is more like an experiment report rather than scientific literature.

Author Response

Thank you for your opinions. Please check the attacted file. 

Reviewer 4 Report

The authors conducted Bead On Plate (BOP) welding tests to determine the range of welding conditions of 6 mm thick 9% nickel steel plates.

(1) Following sentence in abstract and conclusion: "the tendency of penetration  shape was analyzed through a fiber laser Bead on Plate (BOP) experiment for 9% nickel steel with  a thickness of 6 mm and a range of welding conditions for 1-pass laser butt welding of 6 mm thick  9% nickel steel were derived."  In practice, even under the same conditions, the weld geometry & sizes are quite different between BOP welding and butt welding. Thus, BOP welding test results may be  only of some reference meanings, but not for direct usefulness for selecting process parameters in butt weldign cases.

(2) BOP cannot be used alone. It must be used with welding, i.e., BOP welding. Thus, "bead on plate of 6 mm..." in title, BOP test, BOP operation etc. in the text, are not appropriate.

(3) Table 4: for each case, how many times of tests were repeated? Were the measured sizes of welds based on one time measurement or the averaged values of a few times? 

(4) Fig. 4, the length scale cannot be readable.

(5) Figs. 5-7: can you give the error bar for each measurement point?

Author Response

(The authors gave the same response as above.)

Round 2

Reviewer 2 Report

The authors prepared a second version of the manuscript with minimal changes.
I am very sorry, but the discrepancy between the overuse of LNG in the
first part of the manuscript and the absence of LNG in its second part
cannot be resolved by modifying the title of the article. In the end, there was a hint of a novelty in the form of an effort to
replace the existing FCAW welding with fiber laser welding in a specific
application. However, the authors completely resigned from the discussion,
and I insist that the article does not contain scientific conclusions,
only a summary of the experiments performed.

Therefore, in my opinion, the article does not meet the minimum
requirements for publications in the Materials journal.

Author Response

(The authors gave the same response as above.)

Reviewer 3 Report

All my previous concerns have been addressed.

Reviewer 4 Report

All issues have been addressed.